# Methodological Approaches for Risk Assessment of Tobacco and Related Products

**DOI:** 10.3390/toxics10090491

**Published:** 2022-08-24

**Authors:** Yvonne C. M. Staal, Peter M. J. Bos, Reinskje Talhout

**Affiliations:** National Institute for Public Health and the Environment, P.O. Box 1, 3720 BA Bilthoven, The Netherlands

**Keywords:** tobacco products, risk assessment, mixtures

## Abstract

Health risk assessment of tobacco and related products (TRPs) is highly challenging due to the variety in products, even within the product class, the complex mixture of components in the emission and the variety of user behaviour. In this paper, we summarize methods that can be used to assess the health risks associated with the use of TRPs. The choice of methods to be used and the data needed are dependent on the aim. Risk assessment can be used to identify the emission components of highest health concern. Alternatively, risk assessment methods can be used to determine the absolute risk of a TRP, which is the health risk of a product, not related to other products, or to determine the relative risk of a TRP, which is the health risk of a TRP compared to, for example, a cigarette. Generally, health risk assessment can be based on the effects of the complete mixture (whole smoke) or based on the (added) effects of individual components. Data requirements are dependent on the method used, but most methods require substantial data on identity and quantity of components in emissions and on the hazards of these components. Especially for hazards, only limited data are available. Currently, due to a lack of suitable data, quantitative risk assessment methods cannot be used to inform regulation.

## 1. Introduction

Tobacco use is the major cause of premature death worldwide. Each year, about 8 million people die from tobacco-related diseases, including an estimated 1.2 million non-smokers who were exposed to second-hand smoke [1]. Although cigarettes are the most common tobacco product, especially in developed countries, other tobacco products also pose serious health risks. In India, more than 350,000 deaths are attributed to use of chewing and oral tobacco each year [2].

The toxic effects associated with the high mortality rate associated with tobacco consumption are due to carcinogenic and otherwise hazardous tobacco constituents and combustion products. The contributions of individual components to the carcinogenicity of tobacco use have been estimated [3,4], leading to identification of the major carcinogens and ranking of smoke constituents by their potency in inducing tumours. Similar approaches may also be used for cardiovascular and other health risks.

Strategies have been proposed to reduce the exposure of smokers to toxicants, including mandatory limits on the most relevant toxicants in cigarette smoke [5,6,7]. In addition, new tobacco and related products (TRPs) have been developed which have lower quantities of specific toxicants in their emission, such as heated tobacco products (HTPs) and e-cigarettes. These products may also contain other components than cigarettes, such as specific flavourings. We have defined TRPs as all tobacco products and all other products that may be used as alternatives to tobacco products; this includes both nicotine- and non-nicotine-containing products but excludes nicotine replacement products, as such products are not intended for replacing TRP use.

A smoker switching to a product that is potentially less harmful may experience a reduction in health risk, whereas the same product will lead to an increased health risk for a non-smoker compared to no TRP use at all. Quantitative hazard characterization, which includes a dose– or concentration–response relation, will give information on the health impact of a TRP. When information is available on the number of users and their use patterns, such hazard data can be used to obtain information on quantitative health impacts to determine the potential health effects at population level. Ideally, risk assessment of TRPs should be conducted separately for groups of devices or even for individual products [8]. As there is a wide variety in individual puff topography, a wide range of topographies must also be considered in estimating human exposure. This includes using relevant smoking topographies in smoking machines to characterize emissions. Practically, this ideal approach is not feasible since the variation in topographies is huge. A pragmatic solution would be to define extremes in the composition of the emission, using extreme (high and low), but realistic smoking topographies to define ranges for concentration of components in the emissions and whether the composition changes with topography. Such extremes in emission can be used to group TRPs or use scenarios and to select TRPs for a product-specific risk assessment. In such cases, generalization of risk assessment to product classes may be scientifically justified and a more pragmatic way to proceed.

This paper gives an overview of risk assessment methods that can be applied to get insight into the health impact of TRPs. The methods are described with their respective pros and cons when applied to assess the risk of a TRP. With this paper we aim to provide guidance for deciding which risk assessment method is relevant to apply in a specific case based on the information needed, the outcome and the limitations of the method. Risk assessment on population level comes with more challenges, such as the role of marketing in product initiation, addictive potential and attractiveness of the product [9,10]. Our paper focusses on toxicological risk assessment of a product as such and therefore does not discuss these other important aspects, although it should be realized that their role should not be ignored.

## 2. Methods for Quantifying Risk

Health risk assessment of TRP use is generally aimed either at assessing the relative (to another product) or absolute health risk of a TRP or to identify components in the emission that have a relatively large contribution to the TRPs’ health risk. This could be used, for example, to set upper limits for specific constituents. The methods used and the data needed are dependent on the aim. Figure 1 gives an overview of risk assessment methods that can be used for these aims in relation to the data demand. These methods will be briefly discussed.

### 2.1. Evaluation Frameworks

Assessment of the health effects of TRPs could be based on an appropriate evaluation framework. In this approach, expert judgement is used to score aspects of a product in order to identify the most important risks of, for example, drugs [11,12]. Such aspects can be predefined properties of a product, such as composition of a product and user-specific characteristics like quantity of use. Each of these aspects is scored based on expert judgement on a scale running from not harmful to extremely harmful. Altogether this results in identification of the aspects of most concern to health. A non-quantitative evaluation framework has also been developed for tobacco products which summarizes all the factors that may influence the attractiveness, addictiveness and toxicity of a product and can be used to identify knowledge gaps or prioritize research on a specific product [9]. Input for such evaluation frameworks is information on product aspects that influence attractiveness and addictiveness in addition to data on the composition of emissions. These models allow evaluation of a product even when limited data are available but can be used to identify possible health risks. In addition, such models can also be used for product scoring, resulting in a quantitative outcome that can be used to compare health risks of TRPs.

### 2.2. Risk Assessment Based on Individual Components

Information on exposure and the hazard of individual components could be used to estimate the risk of a product as a whole, while ignoring the interaction of components in a mixture. For cigarettes, priority components have been identified based on their hazardous potential [6,13,14]. Compared to tobacco cigarette smoke, e-cigarette emissions contain a lower number of components. However, there may be other components in the emission than known tobacco toxicants, such as flavorants [15]. The data on hazards used in this approach are derived from studies providing information on the relationship between exposure and toxicity, including human epidemiological studies and animal experiments. If this relationship can be quantified sufficiently, safe levels of human exposure can be derived. In emissions from TRPs, the concentrations can be above the safe level of exposure, but the concentration and exposure regime (see Section 3) in emission is not the same as the exposure concentration reaching the lower respiratory tract due to dilution of the air by breathing. This should be accounted for and the final concentration in inhaled air should be used for risk assessment rather than the concentration in the emission. Therefore, information on emission composition and concentrations may be used as an indicator of potential concern or can be used to compare products but not directly for quantifying risks. A method based on health risk evaluation of individual components in order to estimate the risk of a complex mixture may result in underestimates of health risks, as interactive effects among components in the mixture are ignored. To compare the severity of effects of components in TRPs, detailed information is necessary on the relationships between exposure and health effects and how they can be extrapolated to effects in humans.

Below we discuss four methods for risk assessment of individual components, the threshold of toxicological concern, the margin of exposure approach, the hazard-quotient/hazard-index and relative potency approach.

#### 2.2.1. Threshold of Toxicological Concern

One approach to evaluate the potential risk of exposure to complex mixtures is the threshold of toxicological concern (TTC) [16]. In this approach, originally developed for preventing risks, the components of potential toxicological concern in a mixture are identified from structure–activity relations and read-across. TTC values (in µg/person or µg/kg body weight per day) have been defined for three classes (Cramer classes I–III) according to structural elements, but only for oral exposure. Cramer class III indicates the highest health risk and consequently the lowest TTC value [17]. The TTC approach cannot be used to quantify health effects and is only designed to identify components for which there is no or low concern.

The risk of a mixture is then assessed by comparing exposure to each of the components in the mixture, with the appropriate TTC value. This results in identification of components with low or no concern and of components with a potential concern. This approach has been applied to complex mixtures such as botanical extracts [18], flavour complexes [19] and, although intended for oral exposure, it has also been applied to inhaled toxicants [20,21,22]. The TTC method might be used when no hazard data are available for the product as a whole or to identify components of potential concern in complex mixtures. This method does not indicate a risk to health but indicates that further testing is required if a component exceeds a TTC threshold; otherwise, the probability of a health risk is low. Although components below a TTC threshold could in combination result in a health risk, this will be limited in comparison with components exceeding the TTC. The TTC method relies on data of known toxicants to identify a possible health risk, which means that sufficient information of comparable components should be available. For TRPs, this method might be used to identify the components in the emission that potentially pose the highest health risk and to prioritize them for further testing. However, as the TTC method and the respective thresholds are based on oral toxicity data, and use a dose relative to body weight, this is quite different from assessments based on inhalation exposure, in which both concentration and exposure duration are important determinants of toxicity [23]. A TTC for inhaled components should be derived from inhalation toxicity data, among other factors, because toxicity is determined by the specific combination of both the exposure concentration and duration, and not just the (inhaled) dose. This is especially important for TRPs as users are throughout the day regularly exposed to peaks of high concentrations of components. Furthermore, an oral toxicity database does not contain information about adverse effects on the respiratory tract which is an important endpoint in inhalation exposure.

#### 2.2.2. Hazard Quotient and Hazard Index

The hazard index (HI) can be used to estimate the potential risks of a chemical mixture and is defined as the sum of component-specific hazard quotients (HQ) [24,25]. An HQ relates the exposure to a component to a reference value (or limit value) and is calculated as the ratio of exposure and reference value. Assessment factors, for example to correct for differences between animals and humans, have already been considered in these reference values. Therefore, an HQ > 1 indicates a potential health risk for that component. The HI for a mixture of components, as for TRPs, can be calculated as the sum of the HQ for the individual components. An HI > 1 indicates a potential for health risk of exposure to the mixture, and the component(s) which add(s) the most to the HI can be evaluated. However, the HI approach can only be applied to a group of components if the reference levels for the individual components are based on the same health endpoint, i.e., the components need to share a common mechanism of action. For components with a different mechanism of action or a different target organ, subgroups of components should be identified to allow estimation of the effects of a mixture. A further disadvantage may be that reference values may not only reflect the toxicity of a component, since assessment factors applied may not only be health-based but may also be policy-driven or driven by the quality of the database [24].

#### 2.2.3. Margin of Exposure Approach

The margin of exposure (MOE) approach is based on the ratio of the exposure at which no effects occur or the dose at which a predefined adverse effect occurs (e.g., a benchmark dose level) and the exposure level. This approach has been applied to compare components between tobacco products based on potential health risks [26,27,28] and can be used to prioritize components for reduction in tobacco smoke emissions or to assess individual components in the emissions of TRPs. An MOE is calculated for each component from information on hazard and data on inhaled emissions (corrected for inhaled total volume of air, i.e., final concentration in inhaled air). The approach requires relevant hazard and exposure data but does not result in a quantification of the health risks. Its main goal is to determine whether or not an exposure to a specific component is of concern. The magnitude of the margin of exposure is not a measure of risk and can therefore not be used to compare (the chance of) health risks between components. Furthermore, MOEs of individual components cannot be added straightforwardly to estimate the risk of the combination of components [23]. The MOE approach is a pragmatic approach to compare mixtures consisting of the same components while incorporating differences in exposure. Differences between the exposure pattern on which the hazard information is based and that of the TRP user can be weighed in the evaluation of the magnitude of the margin of exposure (see also Figure 3 in [23] on the application of the MOE approach)). For instance, the impact of differences in the exposure of a TRP user (i.e., frequent high peak exposures during a day) and that of a daily 6 h animal experiment on the health outcome needs to be considered.

#### 2.2.4. Comparison of the HI/HQ Approach and the Margin of Exposure Approach

The main advantage of the HI/HQ approach as compared to the MOE approach is that HQs for different components can be added, provided the aforementioned conditions are met, whereas MOEs cannot be added. However, an important difference between the HI/HQ approach and the MOE approach is the comparator, i.e., a reference value or a point of departure (such as a BMD or NOAEL), respectively. It should be realized that for the HQ (and thus for the HI), issues such as the quality of the data and practical feasibility may have been accounted for in the derivation process of the reference value. Also, the point of departure underlying the reference value may not be the optimal point for evaluation of TRPs as, for example, the exposure scenario may be considerably different. Without verification of the derivation of each reference value, the impact of these issues on the outcome remains uncertain. For the MOE approach the best available data for each component can be used, whereas the HI/HQ approach is dependent on the availability of reference values and these should therefore be based on use comparable data for each component. In addition, the MOE approach has more flexibility and possibilities to account for differences between the exposure patterns of the hazard data and that of the user. This is especially important for TRPs as the reference values or limit values are based on exposure conditions that are highly different from TRP exposure scenarios (see Section 3.2).

#### 2.2.5. Relative Potency Approaches

Relative potency approaches are based on expression of the potency of all components in a mixture with similar toxicity in relation to a reference component. This allows addition of the hazards of individual components to estimate total risk. Such approaches have been applied for components with related structures such as polycyclic aromatic hydrocarbons, dioxines and cholinesterase inhibitors (organophosphates and carbamates) [29,30,31,32]. In addition, studies have been conducted to estimate the carcinogenic potency of a tobacco product as a whole and relative to a (reference) tobacco cigarette [3,33]. In this approach, data from carcinogenicity studies are used to determine the carcinogenic potency of every component by using a modelled linear relation between exposure level and the number of tumours induced. The carcinogenic potential can be compared to a reference value of the index component to calculate the relative carcinogenic potency of each component. This is expressed as a Relative Potency Factor (RPF), which is 1 for the index component and can be higher or lower for the other components. The total relative carcinogenic potency of mixtures or aerosols can then be calculated by adding the concentration values for individual components multiplied by their relative potency and comparing the outcome with the toxicological reference value of the index component. This approach is used for components from different chemical classes and is based on the formation of tumours in general, as opposed to being organ-specific. However, components should show similar toxicity, and the mixture components show similar dose–response curves on a log scale (i.e., only differ in potency) and it is assumed the mixture components do not interact (i.e., do not show synergism or antagonism) [32]. Stephens (24) modelled the carcinogenic potency of aerosols from cigarettes, e-cigarettes and HTPs, and comparative modelling approaches have since been refined [3] to determine the relative cancer potency of individual components and product emissions, with confidence intervals. The ratio of cumulative exposure can then be calculated with a probabilistic approach for two products. For HTPs, the ratio of cumulative exposure to selected components was 10–25 times lower than from smoking cigarettes [3]. With relevant information on human dose responses, the change in cumulative exposure can be translated into an associated health impact for each device. This approach was initially used for eight carcinogens that occur in the aerosol of HTP and in cigarette smoke but should be extended to carcinogenic components that are found at higher levels in HTP aerosols than in cigarette smoke. This relative potency approach depends on the availability of either substance- or product-specific data on both emissions and carcinogenicity [34].

### 2.3. Risk Assessment of the Product as a Whole

Hazard assessment of the product as a whole can be done using epidemiological data, in vivo studies or in vitro models, which will give information on (adverse) effects in response to an exposure. Epidemiological studies might be preferred, but also have their limitations. For example, human studies with TRP users often involve former smokers, for which delayed effects of former smoking complicate the hazard assessment of the new TRP. In addition, many TRP users are also dual users (i.e., parallel use with tobacco cigarettes or other tobacco products). Unfortunately, epidemiological studies of the health effects of consistent exclusive e-cigarette use (or other TRP use) without a previous smoking history are difficult to conduct because of the relatively small population of non-former smokers and current e-cigarette users [35]. In addition, there are many confounders for TRP use, such as other life-style factors, as well as social and economic status, which make interpretation of the effect of TRP use challenging. In contrast to epidemiological studies, clinical studies assessing the effect of TRP use in a defined population are less impacted by these confounders.

On the other hand, bioassays in experimental animals may have disadvantages due to interspecies differences and ethical objections, and the results of cellular assays are difficult to translate into effects in humans. In addition, not only should the effects (read-out parameters) be extrapolated to human effects, but the exposure should resemble human exposure. This includes smoking topography and, in the case of a lung model, deposition in the airways. As the exposure in in vitro and in vivo differs largely from human exposure to TRPs, as explained above, such methods can be used for hazard assessment and can provide input for risk assessment, but only in combination with appropriate exposure information to bridge these differences. In vitro and in vivo studies can be used to determine an exposure -elated response, for identifying relevant target organs or modes of action for adverse effects or for determining the human relevancy of an effect.

### 2.4. Possibilities and Limitations of Risk Assessment Methods

The methods described above can be used to compare the health risks of different products. Table 1 summarizes the methods, their data requirements and their applications. The relevancy of applying the methods is determined by the information available. In some methods, a weight-of-evidence approach can be used for data of different quality. All methods for risk quantification also require data on emissions, an indicator of human exposure. It should be noted that all the methods are described to assess health risks of users. Similar methods could be used to assess the risk of bystanders (second or third hand smoke exposure), provided that information is available on their exposure. In such cases, the exposure route may not be limited to the inhalation route of exposure, but oral and dermal exposure should be considered as well.

## 3. Challenges to Quantifying Risk

Quantification of the risks of chemical mixtures is inherently difficult, because of the interaction that may occur between components and because the effects of single components need to be factored in. In the case of TRPs, there are some additional topics that determine the exposure of the user and need to be accounted for: product variation, user-related factors and the complex composition of the emission. There is large variation in the product itself. For example, in heated tobacco differences between sticks of the same brand, and devices used to heat them, will lead to different emission profiles. Such differences will become even larger when one considers not one brand, but an entire product class. To complicate things further, this differs per consumer due to variation in the way the product is used, which affects the identity and quantity of the emission profile and the exposure pattern. Finally, health risk assessment of TRPs involves some complexities due to the complex mixture of components in the emission which may not be constant. These three topics are briefly discussed separately in the next three sections, although it should be noted that these topics may be interrelated.

### 3.1. Product Variation

The TRPs with currently the widest variation in heating devices and fillings (and their combinations) is the e-cigarette or electronic nicotine delivery systems (ENDS). The vapour that is inhaled by a user is dependent on the system itself, the possibilities to adapt the system, the adaptations by the user and the composition of the e-liquid. There are over 20,000 varieties of e-liquids notified in The Netherlands [36]. In the case of refillable e-cigarettes, endless combinations of devices and e-liquids can be made, which allow the user to adapt and choose the settings and the e-liquids he/she prefers. Some of these product variations are also applicable to other TRPs than e-cigarettes, such as different flavours of tobacco stick for HTPs. Variations in the product lead to changes in presence as well as the quantity of the components in emission.

### 3.2. User-Related Factors

A major complexity in using exposure information for risk assessment is that the exposure scenario that is used to determine hazard is substantially different between hazard assessment studies and TRP users. Inhalation studies are preferred over oral studies, since the exposure route is more relevant for TRPs and health risks will be related to the specific inhalation exposure characteristics (concentration, duration, frequency). However, exposure in experimental inhalation studies in animals is generally for 6 h/day, 5 days/week, which is not representative for the use of TRPs, as TRP use generally results in irregular peak exposure for 7 days/week. Therefore, studies in experimental animals may not provide meaningful results for assessing the risks that the complex exposure scenario of TRPs poses to humans [23]. The development and use of alternative models, such as cell models, are increasing rapidly, and may help to apply more relevant exposure scenarios in the near future [37,38]. Exposure scenarios for cell models would be based on local concentrations at the site of the cell and allow more rapid assessment of different exposure scenarios in relation to their effect. In vitro read-outs will, however, need to be extrapolated on the basis of effects at the organism level [39,40].

### 3.3. Complex Mixture of Components in the Emission

Tobacco smoke consists of a mixture of over 7000 chemicals, while the emission of many TRPs is less complex. The complexity of the product emissions is dependent on the product itself (as mentioned in Section 3.1) and the user (as mentioned in Section 3.2). For risk assessment methods that rely on the effects of individual components, the components in emissions must be characterized and quantified in order to assess the risk of these products [41]. Unfortunately, information on ingredients (contents) alone is insufficient, as they may not completely transfer into the emissions, they may degrade or burn during aerosolization or as components in the emissions may originate from the device (such as metals). Information on the chemical composition of the emissions is necessary to identify the components to which users are exposed. This mixture of components in the emissions varies both in presence and in quantity for the individual constituents. As this mixture is dependent on the user behaviour, this makes a risk assessment of TRP emission specific for a combination of a product and user or, to reduce complexity, a user group. Generalization of risk assessment to a group of products, for example e-cigarettes, relies on assumptions about limited variation or representative product choice, which are difficult to substantiate, and it is difficult to define their impact on health risk. Insight into the drivers of the variations in emission will help to group products according to their emissions, which can be used to substantiate grouping of TRPs for risk assessment purposes, to, ultimately, assess the risk of this group of TRPs.

Some work has been conducted on the toxicological effects of mixtures [42] to determine whether the effect of the mixture was different from those of the sum of the individual components [43]. For such purposes, components are often classified according to their target organ and their mechanism of action. Most mixture assessments have focused on binary mixtures, but risk assessment of the complex emissions of TRPs is even more complicated and is similar to the assessment of other complex mixtures, such as petroleum-derived products and air pollution [44,45].

To add to this complexity, TRP emissions are dynamic. Emissions cool as they pass to the exit of the device or the cigarette on their way to the respiratory tract and get humidified along the way, resulting in condensation of volatile components, agglomeration of particles, reactions of components with each other (aerosol aging) or binding to water in the humidified air. These processes occur simultaneously and determine local deposition in the airways, which can result in high doses at specific locations in the airways, which could have site-specific adverse effects. Models are being developed to estimate airway deposition of tobacco smoke and e-cigarette emissions to allow assessment of local dose; however, most models focus on a few components, not on complete emissions [46,47].

A summary of the factors affecting the exposure of a user is shown in Table 2.

## 4. Discussion

### 4.1. Overview and Applications

This paper provides an overview of methods that can be used to assess the health risks associated with the use of TRPs. Several models are available that could assess the risk of mixtures in TRPs, although most address carcinogenic effects. The methods described in this paper can be used for assessing the risks of TRPs, each aimed at answering a different question (Table 1). Moreover, probably more than one model will be required for a full assessment, which is dependent on the regulatory or scientific question to be answered. This question includes, amongst others, the group that is exposed (smokers, non-smokers or bystanders, for example). Methods based on the risk associated with components in emissions can be used to obtain an indication of the absolute or relative health risk of a product. At this time, not enough scientific data are available to make full health risk assessments of a TRP, but whether that is needed depends on the aim of performing TRP risk assessment. When more hazard information is available, only chemical analysis of the emissions of a novel TRP would be required, which, combined with models of deposition and risk assessment, would allow determination of the health effects.

### 4.2. Risk Characterization

Risk characterization requires information on the relation between actual human exposure and the occurrence of adverse effects. Such a relation is important to validate the methods for TRP risk assessment, and to ultimately apply risk assessment methods for novel TRPs prior to their market launch, when only limited information is available. A causal relation between TRP use and acute effects (short-term health risk) is generally easier to identify than the effects on the longer term, as the time between exposure and effect is short. In many cases, when users stop using the product the adverse effects may be mitigated. Assessment of the health risk of TRPs would benefit from data on health effects in long-time users; unfortunately, such data are not yet available, as novel TRPs have not been available for the time necessary to develop chronic health effects such as cancer. In addition, current TRP users are often former smokers. Thus, if a user develops a disease, it may be a delayed effect of smoking and not necessarily related to TRP use. The most robust data for assessing health risk would be for TRP users who are not former smokers and not dual users. The lack of long-term data and of information on non-smokers may change over time as the products remain on the market for longer. This is exactly why the methods to characterize TRP risk described in this paper are needed, since these can be applied before products are launched into the market.

### 4.3. Risks at Population Level

A quantified health risk of a TRP can be used to provide information on health risk at a population level of that TRP, when combining this with information on the number of users and the quantity of the TRP used. Although this has not yet been applied in practice, the feasibility of modelling population health effects has been explored [48]. When quantitative information on health risk and product use across the population is available, the health impact of TRPs in smokers, non-smokers and former smokers can be estimated when monitoring the popularity of the TRP (number of users) and how the TRP is used. The outcomes can be used to inform legislative measures to, for example, regulate contents and emissions or establish a basis for public education. It should be noted that quantification of health risks is not a static outcome but remains an estimation based on the available knowledge and is always influenced by the user and frequency of use. Information on novel TRPs is increasing, as is, probably even more important for e-cigarettes, the wide variety of devices, user settings and e-liquids, which will influence health risks.

### 4.4. Implications for Regulation

The risk assessment approaches described in this paper could inform policymakers on the health effects of a product or could be considered for use in regulation. However, there is insufficient data to reliably quantify the health risk of TRPs, and there is no uniformly used method to quantify risks of complex mixtures. Whether such information is needed, also depends on the regulatory aim, as most tobacco product legislations do not require detailed information on absolute risk of a TRP. As a first step, the conceptual model can be used to identify whether there are any health concerns to be expected, but the decision to apply subsequent models is dependent on the question that needs to be answered and the data that is available.

### 4.5. Recommendations

Development of risk assessment models should continue and, at some point, they should be validated with human data. Models of airway deposition should also be developed for application in risk assessment, as this is a crucial step between emission quantification and hazard characterization. From a scientific perspective, further development and ultimately implementation is currently limited by lack of data, which also implies that the models cannot yet be validated with data on human use. For a meaningful application of quantitative methods of risk assessment, data should be collected on the emissions, toxicity, use and effects of TRPs on exposed populations. Characterization of toxicants should include non-targeted screening approaches to identify product-specific components that are not usually measured in tobacco smoke.

It is recommended to evaluate the suitability of a framework published by Meek et al. [49] for combined exposures for the risk assessment of TRPs, in which the methods discussed come together. As follow-up of a WHO/IPCS Workshop on Aggregate/Cumulative risk assessment, Meek et al. [49] published a framework designed to aid in identifying priorities for risk management for exposure scenarios with combined exposures. Evaluation is done using a tiered approach which combines exposure assessment and hazard assessment. Along the evaluation, more refined tools are used. At any tier, the evaluation is made by calculating an MOE and the outcome of the analysis can be risk management, no further action, or further assessment. The assessment stops if an adequate assessment can be made. The framework helps to identify potential data gaps that need to be filled before the step to a next higher tier can be made. In addition to the MOE approach, the other methods also discussed in the present paper can be used in this framework, including the TTC approach, the Hazard Quotient and Hazard Index, and the use of relative potency factors, as is illustrated by the two example cases described in the paper by Meek et al. This framework, therefore, may provide useful guidance for the evaluation of combined exposure to multiple chemicals, as occurs when using TRPs.

From a regulatory perspective, these risk assessment methods can be selected based on regulatory needs, and based on these needs, address the requirements for data. These data requirements could be provided by the producers of TRPs, while following quality criteria [34] and using human-relevant scenarios to ensure its reliability and applicability. Such data would not only benefit risk assessment of TRPs but may also help to select ingredients, emissions and technical features that have the strongest contribution to health risks.

## 5. Conclusions

Several approaches have been used to quantify the health risk of tobacco products, either the absolute risk or that relative to a tobacco cigarette. The HI and RPF approaches may be used for quantification of health risk, provided that sufficient and relevant hazard and exposure data is available. None of the methods are ready to be used in regulation yet due to a lack of relevant data on hazard and exposure, but also due to a variety of regulatory needs and wishes. Nevertheless, application of these methods may be possible in due time.

## Figures and Tables

**Figure 1 toxics-10-00491-f001:**
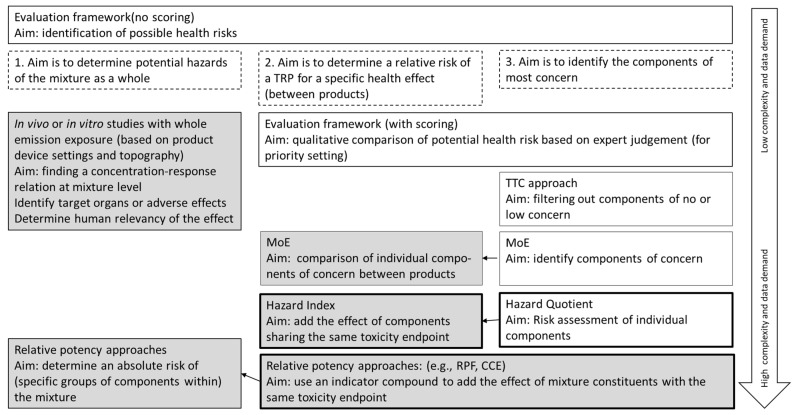
Overview of risk assessment methods for health risk assessment of TRPs. The choice of the method is dependent on the aim. Dashed boxes: the three different aims of the risk assessment methods. The methods that can be used for this aim are in the same column. White boxes: Methods resulting in an assessment for the individual compounds in the emission. Grey boxes: methods resulting in an assessment for the mixture of components. Interaction between components is not considered, except for experimental studies with whole emission. All methods, except experimental studies with whole smoke, are dependent on the available data on emission composition and on hazard for individual components. Black-lined boxes: these methods allow quantification of risk of single components. The arrows between boxes indicate a follow-up of that method; for example, the MoE approach first needs to be applied to identify components of concern before it can be used to compare these components between products. The arrow on the right indicates the complexity and data requirements of the methods.

**Table 1 toxics-10-00491-t001:** Main limitations and advantages of each method for quantifying the health risk of TRPs.

	Potential Application for TRPs	Main Limitations	Main Advantages
Evaluation frameworks (with or without scoring)	Qualitative health risk assessment based on scores, can be used for setting priorities	Most subjective method No quantification of risks	Requires limited data; more data will improve outcomes
Threshold of toxicological concern (TTC)	Identification of components for further assessment/testing	Cannot assess risk of complete product. No quantification of risks	Identification of components of no concern
Hazard quotient (HQ)/Hazard index (HI)	Health risk assessment based on available data Health risk assessment of groups of components sharing the same toxicity endpoint	High data requirement. Only for groups with reference value based on similar toxicity endpoint Assessment factors may be based on non-scientific considerations	Considers target organ in the evaluation
Margin of exposure approach (MoE)	Identification of risks of components of concern Comparison between products on risks from individual components	High data requirement Cannot sum risks of different substances	Identification of individual components of (potential) concern
Relative potency approaches	Health risk assessment based on total risk of groups of components sharing the same toxicological endpoint Comparison between products based on groups of components	High data requirement for all components within a group. Components should share the same toxicological endpoint	Allows comparison of risks between products for groups of components
In vivo or in vitro studies with whole emission exposure	Hazard assessment based on dose–response data of mixture as a whole	Extensive testing required and extrapolation of exposure and results to humans Only information on one composition	Does not require data on emissions or hazard of individual components as the model is exposed to the emission as a whole Includes agonistic and antagonistic effects of all components

**Table 2 toxics-10-00491-t002:** Factors that determine exposure and deposition in the respiratory tract of TRP emissions, while using the e-cigarette as an example.

	Factor	Effect on
Product-related	Settings of the device	Identity and quantity of components in emission, particle size distribution
Product-related	Product itself (such as brand)	Identity and quantity of components in emission, particle size distribution
User-related	Topography	Identity and quantity of components in the emission, user exposure
User-related	Number of items consumed per day	Quantity inhaled of each component, user exposure
User-related	Breathing volume	Quantity of air inhaled with a puff dilutes the emission and therefore determines the concentrations inhaled
Complex mixtures	Burning and degradation	Identity and quantity of components in emission
Complex mixtures	Emissions from other sources, such as the device	Identity and quantity of components in emission
Complex mixtures	Aerosol aging, humidification in the airways	Particle size distribution

## Data Availability

Not applicable.

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
