# Peer review of "Methodological Approaches for Risk Assessment of Tobacco and Related Products"

_toxics, 2022, doi:10.3390/toxics10090491_

Round 1
Reviewer 1 Report
The article touches upon an interesting medicine issue. The paper is comprehensible and does not feature any repetitions. The research has been constructed appropriately. The findings have been presented clearly and the discussion is related to research results. The literature has been collated appropriately.
Author Response
Thank you very much for your positive review of our manuscript.
Reviewer 2 Report
This paper discusses different methods for risk assessment of tobacco and related products. The manuscript provides valuable information on different risk assessment methods for tobacco products.
I have a few comments below:
1. There are some studies on risk assessment for tobacco regulation. This is an example: https://doi.org/10.18001/TRS.5.1.4.
This work provides a review of the risk assessment framework developed by the National Research Council and typically applied by federal regulatory agencies. I feel that it may be a good idea to discuss such frameworks in the presented work.
2. There are some new studies aiming at the risk assessment of cigarette smoking. Here is a recent one: https://doi.org/10.3390/ijerph19063746.
The authors are invited to consider discussing such studies as examples of risk assessment methods application.
3. Regarding exposure to tobacco products, the route of exposure should be considered. That can be true that the primary way of exposure to tobacco products is through inhalation. However, there are some pieces of evidence indicating that exposure could be also through dermal and oral. For example, I invite the authors to consider this published work (https://doi.org/10.1136/tobaccocontrol-2016-053602) and consider a short discussion on considering the "Challenges of quantifying risk" section.
4. I am wondering if it is possible to categorize the subjects of risk assessment and discuss which method can be the best risk assessment method for a specific subject. For example, what method of risk assessment can be the best for smokers, and what can be the best when considering second or thirdhand exposure? The applicability of each method may be discussed in more depth by considering such issues.
Overall, I found this manuscript to be very informative. I wish the comments above be useful to the authors.
Author Response
We would like to thank you for your review and valuable comments on our manuscript.
- Thank you for this comment and reference. We acknowledge the inherent challenges that come with risk assessment of tobacco products, specifically the role of marketing and promotion, packaging/branding, price, social influences, consumer perception, and addiction as mentioned in the paper, on the health effects on population level. We have added a sentence to the last paragraph of the introduction to address this point, but also to clarify that our focus is on toxicological risk assessment of the product as such.
- This reference addresses biomarkers of exposure to second and third hand smoke. We agree that this is a nice example of application of the HI method for risk assessment of bystanders, but as this is based on biomarkers of exposure and not the exposure itself, we think that it would not suit as an example in our case.
- This paper also addresses third hand smoke. We had briefly touched upon exposure of bystanders and now extended this to address possible different routes of exposure for second and third hand smoking.
- Thanks for this interesting question. We think it should not only be the subgroup of the population that should be considered, but also the (regulatory) aim (now addressed in the discussion). There could many questions that can be answered with the methods we describe and it is not feasible to address all these questions as it is impossible to be complete. This is actually a point that should be addressed in a more complex risk assessment approach as we have described earlier and also approached in the reference you mentioned. For such an approach it is especially important to consider attractiveness and addictiveness of products, together with the number of people using different TRP.
Reviewer 3 Report
Thank you very much for the opportunity to review this research. The authors undertook the difficult task of discussing and evaluating methods for assessing the health impact of nicotine products. They also highlighted the advantages and disadvantages of these methods.
Below you will find a few of my comments on the text.
Line 11 - Alternatively, Risk - "Risk"- lowercase letter. Please, change.
Line 19 - regulation.. double fullstop - please change.
Line 44 - "A smoker switching to a product that is potentially less harmful may experience a 44 reduction in health risk, whereas the same product will lead to an increased health risk 45 for a non-smoker relative to no TRP-use at all". - Did you mean "in relation to" instead of "relative to"? Please, explain.
Line 240 - "Stephens (24) modelled the carcinogenic 240 potency of aerosols from cigarettes, e-cigarettes and HTPs, and comparative modelling 241 approaches have since been refined [3] to determine the relative cancer potency of indi- 242 vidual components and product emissions, with confidence intervals". - Why are these references in italics?
Line 338 - The Netherlands [35]. - doubled space. The same for line 342.
Line 350 - (concentration, duration, frequency).. - doubled dot
Line 502 - and exposure data is available None - lack of the fullstop.
Author Response
Thanks for your feedback on our manuscript.
The corrections in lines 11, 19, 240, 338, 342, 350 and 502 have been applied.
Line 44: we see this may be confusion. We have changed ‘relative to’ into ‘compared to’.